# Precipitation and Moisture Transport of the 2021 Shimokita Heavy Precipitation: A Transformed Extratropical Cyclone from Typhoon#9

**Akiyo Yatagai [1],\* and Shogo Saruta [2]**

1   Graduate School of Science and Technology, Hirosaki University, Hirosaki 036-8561, Japan
2   Faculty of Science and Technology, Hirosaki University, Hirosaki 036-8561, Japan
*   Correspondence: yatagai@hirosaki-u.ac.jp; Tel.: +81-172-39-3685

**Abstract:** This study examines the heavy rainfall event that occurred in the Shimokita Peninsula, Japan, on 9–10 August 2021, resulting from an extra-tropical cyclone that developed from Typhoon#9 (EC9). The objective of this study is to elucidate the relationship between moisture transport and heavy rainfall and to verify the role of EC9. The authors created intensive hourly precipitation data over the Aomori Prefecture and analyzed them together with moisture fields. In most locations where the landslide disaster occurred, there were two precipitation peaks: at 9 UTC and 18 UTC on 9 August. The wind shear was strong from the lower to the upper troposphere with easterly winds in the lower troposphere and warm moist air from south for the first peak. A strong horizontal gradient of equivalent potential temperature, a northerly in lower troposphere, and moisture convergence over Shimokita Peninsula indicate the existence of the stationary front for the latter peak (18 UTC). The heavy precipitation and moisture convergence that caused the Shimokita event were identified by the stationary front of EC9 around the latter peak (15 UTC of 9th–06 UTC of 10 August). The precipitation distribution, which has a precipitation peak northeast of the EC center, is a typical typhoon-turned extratropical cyclone (EC) precipitation distribution.

**Keywords:** precipitation; typhoon; water vapor; landslide; transformed TC

## 1. Introduction

From 9 to 10 August 2021 (JST), an extra-tropical cyclone (EC) that transformed from Typhoon#9 moved northeastward over the Sea of Japan and passed through the Tohoku region. Some areas of the Shimokita Peninsula, Aomori Prefecture (Figure 1), experienced extremely heavy rainfall from the evening of 9 August to the morning of 10 August [1]. This heavy rainfall, hereafter Shimokita event, caused many landslides in Kazamaura Village and Mutsu City [2], resulting in traffic disruptions owing to collapsed bridges and damage to buildings.

The western Pacific has the most tropical cyclone (TC) transits, and TC precipitation is the largest in the northwest Pacific among the world's oceans [3]. A climatological view of TC precipitation has been made with satellite observations such as Tropical Rainfall Measuring Mission (TRMM) [3,4]. The land-falling precipitation associated with TCs, which are important in terms of disaster prevention, have been clarified using radar networks (e.g., [5–7]) as well as rain gauge based accurate precipitation data sets [8].

The area where this Shimokita event occurred is at mid-to-high latitude (about 40 N), where TCs rarely make landfall. Heavy rainfall caused by typhoon-turned-ECs is more likely to occur in the northern Tohoku region than in areas farther south, but there are few studies on this topic. There are theoretical studies and classifications on the transformation of a typhoon into an EC using satellites [8]. However, large differences among the cases of typhoon-turned-ECs are considered [9–11], so case studies using quantitative data are essential. In addition, the precipitation associated with transformed ECs has not yet

been reported statistically or comprehensively. Therefore, it is important to clarify the precipitation distribution of extratropical cyclones that caused disasters for the validation of numerical models and for future disaster prevention.

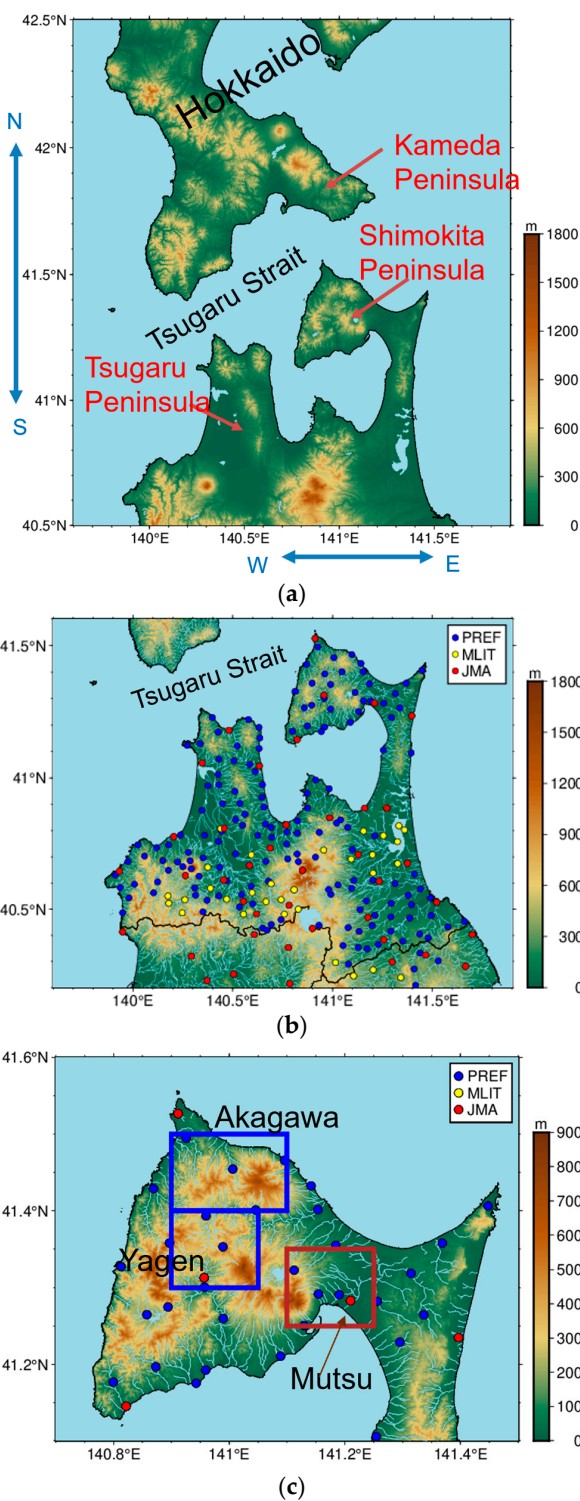

**Figure 1.** (**a**) Geography of the target area. Color indicates elevation (m), and color scale is the same as that in (**b**). Two blue arrows indicate the range of cross sections shown later figure. (**b**) Elevation and rain gauge stations over Aomori Prefecture and (**c**) Shimokita Peninsula. Blue dots indicate Aomori Prefecture's rain gauge; yellow dots indicate the Ministry of Land's rain gauge; and red dots represent that of JMA's. Blue boxes in (**c**) indicate the Akagawa River basin and Yagen where landslide damage was severe.

The factors that caused the heavy rainfall of the Shimokita event were reported to be the northward expansion of a low-pressure system and front (Figure 2), increased moisture flux due to enhanced easterly winds, convergence at the eastern slopes of mountains, and southerly winds 5000 m above the ground [1]. A sensitivity experiment using the Japan Meteorological Agency Non-Hydrostatic Model (JMA-NHM) [12] was conducted for 9 August, but it was focused on the first peak (around 9 UTC) of precipitation (Figure 3b). They discussed the cause of a line-shaped precipitation band running north–south from the Shimokita Peninsula to the Kameda Peninsula in Hokkaido (cf. Figure 1a).

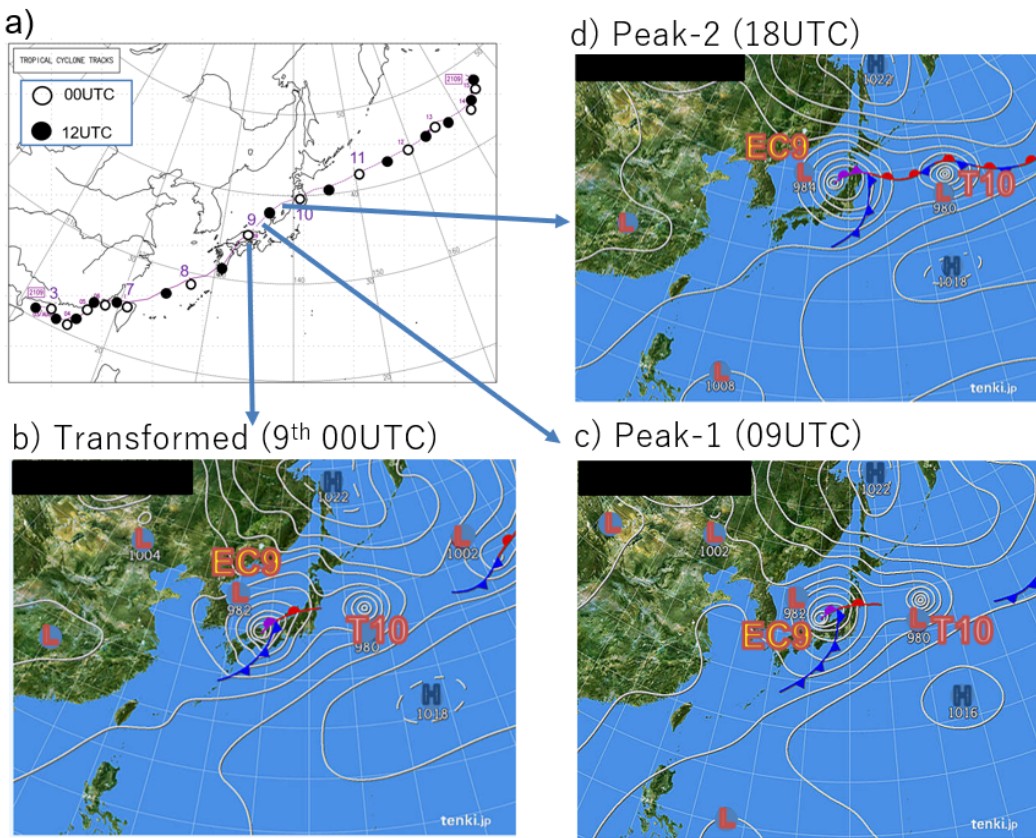

**Figure 2.** (**a**) Path of Typhoon#9 in 2021. Open circles indicate the TC/EC center location at 9 AM JST (=00 UTC), and closed circles indicate that of 9 PM JST (=12 UTC). The numbers with small subscripts indicate the date (e.g., "10" indicates 10 August). The best track map is obtained from JMA Tropical Cyclone Tracks (https://www.data.jma.go.jp/yoho/typhoon/route_map/bstv2021.html (accessed on 31 March 2023). (**b**) Weather chart at 9 JST (00 UTC) on 9 August when the TC9 transformed to an extratropical cyclone (EC9). Processed from JWA tenki.jp "Past weather charts, August 2021" (https://tenki.jp/past/2021/08/chart/) (accessed on 31 March 2023). (**c**) Same with (**b**) but at 9 UTC. (**d**) Same with (**b**) but at 18 UTC.

However, over Shimokita Peninsula, more precipitation was observed at the second peak (around 18 UTC) than that of the first peak. Since the Shimokita event was a record-breaking event, abundant moisture should be converged. The relationship between precipitation and moisture transport from a larger-scale perspective, such as the role of an extratropical cyclone that was transformed from a typhoon (hereafter, we call "typhoon-turned-EC"), has not yet been reported. An accurate assessment of disaster-causing precipitation and regional moisture transport caused by typhoon-turned-ECs would help improve numerical forecasts and disaster prevention.

Hence, this study aimed to clarify the relationship between moisture transport and heavy rainfall around Aomori Prefecture for the Shimokita event that occurred from 9 to 10 August 2021 and to confirm the contribution of Typhoon#9. Yatagai et al. (2019;

henceforth Y19, [13]) discussed moisture transport for the 2018 torrential rainfall event in western Japan by producing intensive data by the scheme of Asian Precipitation—Highly Resolved Observational Data Integration Towards Evaluation of the Water Resources (APHRODITE) for Japan (APHRO_JP, [14]). We performed a similar analysis for this Shimokita event.

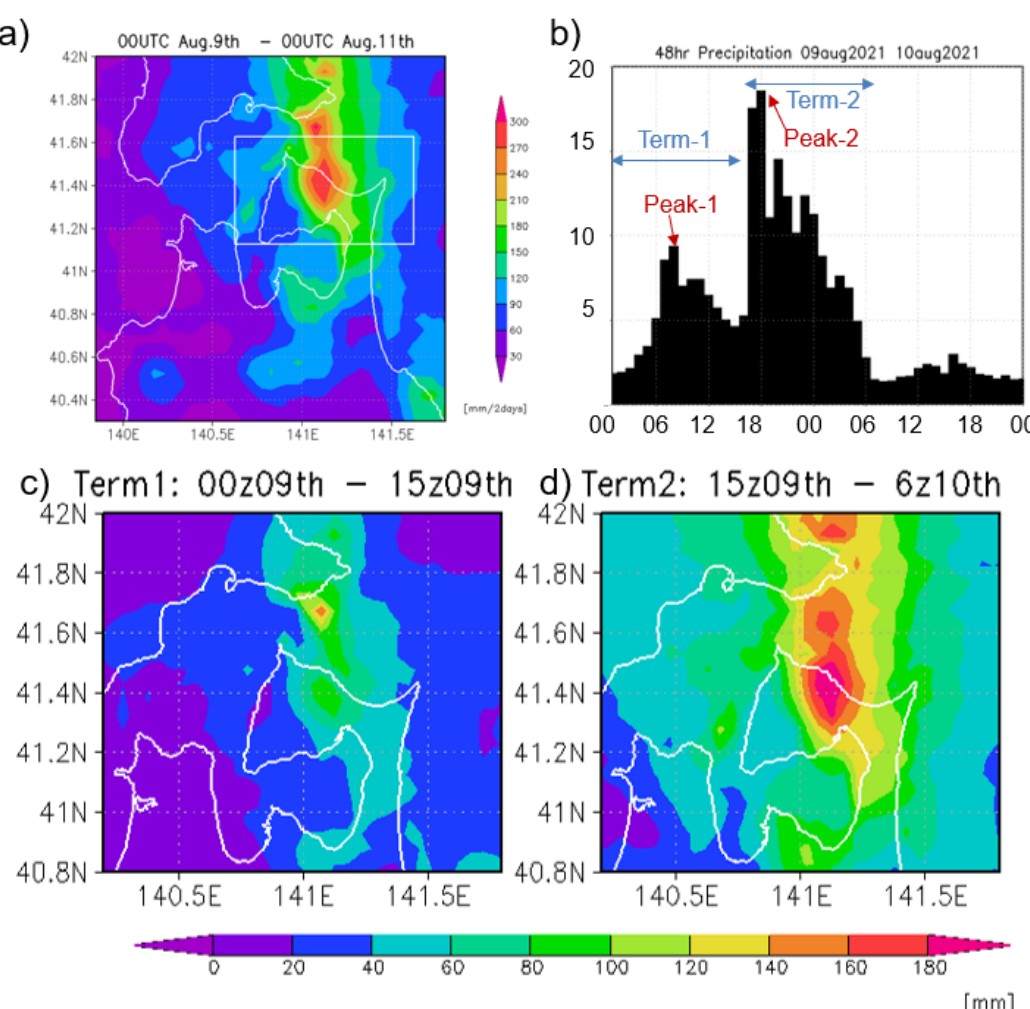

**Figure 3.** (**a**) The 48 h precipitation (APHRO_RA) distribution for the target area (mm/48 h). (**b**) Time series of the average precipitation over a white box designated in (**a**). Two peaks (9 UTC, 18 UTC) and two terms are marked. (**c**) Total precipitation distribution for Term 1 (mm/h) over the Shimokita Peninsula. (**d**) Same as (**c**), but for Term 2 (mm/h).

## 2. Data and Methods

Because this Shimokita event covers 9–10 August 2021, the data used in this study were for this period unless otherwise noted. The typhoon track and weather maps of the target events are shown in Figure 2. The Japan Meteorological Agency (JMA) reported that the Typhoon#9 had turned into an extratropical cyclone at 00 UTC on 9 August [15]. JMA identified the completion of transformation to an extratropical cyclone according to the WMO guidelines [9,11,16], though it is not easy to judge by satellite cloud images (Kitabatake 2013). Here, based on the JMA announcement [15], we assume that the cyclone became an extratropical cyclone on 9 August at 00 UTC. Typhoon#9 is denoted as TC9, and after turning into an EC cyclone, it is denoted as EC9.

## 2.1. Precipitation Data

In addition to AMeDAS, the Automated Meteorological Data Acquisition System (AMeDAS) rain gauge, data from the Aomori Prefecture and the Ministry of Land, Infrastructure, Transport, and Tourism (MLIT) were incorporated for the Aomori Prefecture (Figure 1). The algorithm described by Y19 was adopted, except for the station value conservation (SVC) option. Calculations were performed for the entire Japanese region with a spatial resolution of 0.05° × 0.05° and a temporal interval of 1 h. These data are referred to as APHRO_JPh.

JMA Radar AMeDAS (RA) precipitation data were used for the sea because respective measurements were not available. These combined data are termed APHRO_RA. APHRO_JPh and APHRO_RA refer to 1 h of precipitation from the previous hour to the designated time.

## 2.2. Atmospheric Data

To depict the local moisture transport, equivalent potential temperature, and wind fields, the three-hourly mesoscale model outputs (MSM) released by JMA were utilized. Details of the calculation are the same with that of Y19. The horizontal resolution of MSM (upper atmosphere) is 0.125 degree (longitudinal)/0.1 degree (latitudinal) grid.

For the atmospheric circulation fields and large-scale moisture analysis, the fifth generation of the European Centre for Medium-Range Weather Forecasts (ECMWF) Atmospheric Reanalysis of the Global Climate (ERA5) reanalysis [17] was utilized. The horizontal resolution of ERA5 is 0.25 degree/0.25 degree grid.

Both MSM and ERA5 were used in this study for the following reasons:

MSM has a higher horizontal resolution, which is more reliable when looking at the relationship between water vapor transport in the lower troposphere and the topography around Shimokita Peninsula. Although it is simulated by a non-hydrostatic model, precipitation is forecast and not adjusted by observations. In addition, the MSM data do not include information on humidity above 300 hPa, and the range of the MSM data set is small enough to cover the moisture transport associated with EC9.

In contrast, ERA5 adopts a four-dimensional variational method (4d-var) to make reanalysis data, which assimilates observations of the future for each analysis time as well as that of the past. As a result, for example, diurnal variations in precipitation are well reproduced. Therefore, ERA5 has advantages for representing the location and timing of water vapor transport over a wide area associated with TC9 and EC9.

We calculated the equivalent potential temperature ($\theta_e$) from MSM data using the following equation:

$$\theta_e = \theta_d \exp\left(\frac{L w_s}{C_p T}\right) \tag{1}$$

where $\theta_d$ is the potential temperature of dry atmosphere, $L$ is the latent heat of condensation, $w_s$ is the saturated mixing ratio of water vapor, $C_p$ is the constant pressure specific heat, and $T$ is the temperature.

We compute moisture flux as follows: We use the specific humidity, $q$, and horizontal wind vector $(u, v)$ at each pressure level. Specific humidity is often shown with units (g/kg); however, we take the original units (kg/kg) for calculation, so that the unit of the horizontal moisture flux vector $(qu, qv)$ is (m s$^{-1}$). Then, the unit of moisture flux divergence at each pressure level, $div(qu, qv)$, namely $\nabla \cdot q\vec{V}$ is (s$^{-1}$).

The amount of water vapor contained in a unit area column of air is given by the following expression:

$$W = \frac{1}{g} \int_{p0}^{p1} q \, dp \tag{2}$$

where $g$ is the gravitational acceleration. If we integrate from the earth's surface to the top of the atmosphere, $W$ is called precipitable water, and the unit is (kg·m$^{-2}$) or (mm).

In this study, we integrate $q$, which is integrated for lower troposphere ($p0 = 700$ hPa, $p1 = surface$) and mid troposphere ($p0 = 400$ hPa, $p1 = 650$ hpa).

By integrating the horizontal transport of water vapor with respect to pressure, the following aerial moisture flow $\overrightarrow{QV}$ is obtained:

$$\overrightarrow{QV} = \frac{1}{g} \int_{p0}^{p1} q \overrightarrow{V} \, dp \tag{3}$$

The zonal and meridional components of $\overrightarrow{QV}$ are given by

$$Q_\lambda = \frac{1}{g} \int_{p0}^{p1} qu \, dp, \tag{4}$$

$$Q_\phi = \frac{1}{g} \int_{p0}^{p1} qv \, dp \tag{5}$$

The units of $Q_\lambda$, $Q_\phi$ are (kg m$^{-1}$s$^{-1}$) or (mm ms$^{-1}$). Then, a divergence of $\overrightarrow{QV}$, namely, $\nabla \cdot (Q_\lambda, Q_\phi)$, has the unit of (kg m$^{-2}$ s$^{-1}$). If we multiply the seconds in one hour, 3600 s, with this amount, then we can obtain moisture divergence in one hour (kg m$^{-2}$ h$^{-1}$), and it is the same unit as that of hourly precipitation (mm/h).

*2.3. TC Precipitation Composites*

For reference, a composite figure from the synthetic precipitation database associated with tropical cyclones was created. One-hour precipitation data on a 0.1° grid were created using RA data for Japan and Global Precipitation Measurement Integrated Multi-satellitE Retrievals for GPM (GPM IMERG) Final Precipitation data for the rest of the land and sea. The precipitation distribution was analyzed for tropical cyclones (TCs) that occurred in the northwest Pacific (0° S–60° N, 100° E–180° E) from July to October from 2015 to 2019 by compositing the center location data of the 95 TCs. We treated precipitation within a 500 km radius as TC precipitation, and we composite them according to the angle. Detail is written in [18].

Furthermore, for the Shimokita event period, EC9 composit precipitation is created by APHRO_JPh over the Japanese land area (described in Section 2.1) and GPM IMERGE for the sea and land area other than Japan.

**3. Results and Discussion**

*3.1. Precipitation Distribution and Time Series*

The spatial distribution of APHRO_RA for the 2 days (9–10 August 2021; UTC) is shown in Figure 3a, and the areal averaged hourly time series is displayed in Figure 3b. As noted, there were two peaks at 9 UTC (18 JST) and 18 UTC on 9 August (9 JST on 10 August). Precipitation weakened at 15 UTC. Hence, we termed 0–15 UTC Term 1, while 15 UTC of 9 August to 6 UTC of August 10 is defined as Term 2, as shown in Figure 3b. The areal averaged precipitation time series shown in Figure 3c are shown in Figure S2. Figure 3c,d show the precipitation distribution for Term 1 and Term 2, respectively. Heavy rainfall of the Shimokita event occurred in the area that was not directly observed by JMA's rain gauge (red dots in Figure 1b). Instead, Aomori prefecture's rain gauge (blue dots in Figure 1b) observed, which enabled the successful representation of precipitation by APHRO_JPh.

At the peak during Term 1 (peak 1), EC9 was located near the Noto Peninsula (135.7° E, 37.3° N), while at peak 2 during Term 2, EC9 was located slightly northward over the Sea of Japan (138.2° E, 39.1° N). The latter indicated that the stationary front extended and moved northward (see Figure 2). The location of the landslide disaster (Akagawa, Yagen) in Shimokita is around 141° E, 41.5° N. We plot the locations onto the composited EC precipitation chart in the next section.

### 3.2. Composited TC/EC Precipitation

The TC/EC precipitation described in Section 2.3 is shown in Figure 4 and in Figure S1. TCs were classified into five categories according to maximum wind speed: tropical depression (TD), tropical storm (TS, ≥17 m/s), severe tropical storm (STS, ≥25 m/s), typhoon (TY, ≥33 m/s), and extratropical cyclone (EC). Figure 4 shows composited precipitation distribution (mm/h) for (a) TD, (b) TY, and (c) EC. The areal mean precipitation is the largest TY and smallest at EC among the five categories (Figure S1). The TC precipitation amount increases (TD < TS < STS < TY) as the TC intensifies, i.e., as the wind speed increases, which is consistent with previous studies [4].

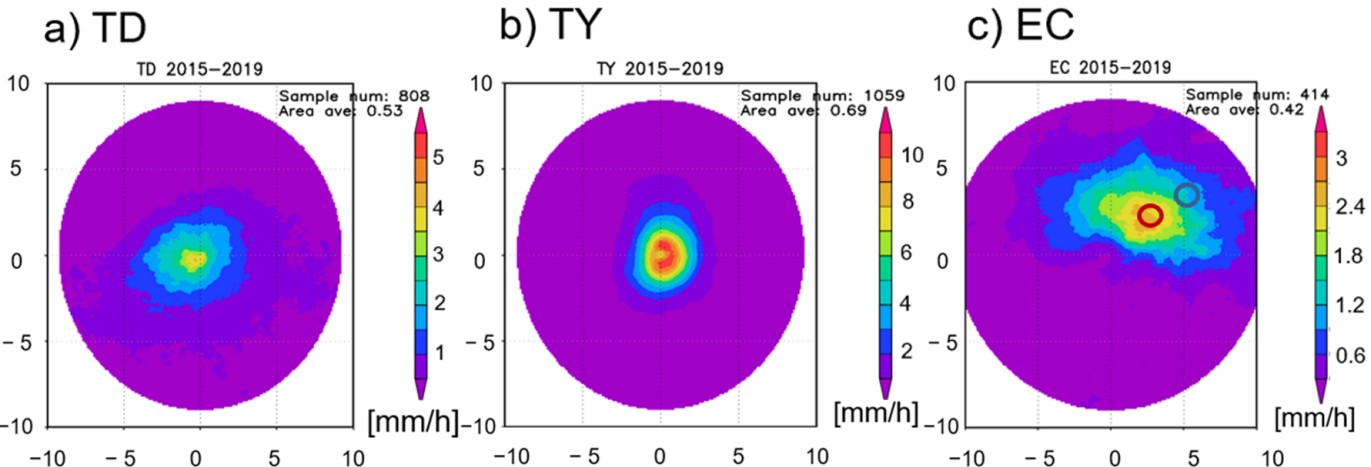

**Figure 4.** Composited precipitation distribution (mm/h) for (**a**) TD, (**b**) TY, and (**c**) EC. Hourly precipitation patterns are composited according to the angle from the TC center. Labels at X and Y axes indicate angle (degree). Blue and red circles in (**c**) indicate relative location of Shimokita from the EC center at peak -1 and peak -2, respectively. The number of samples (N) and areal mean precipitation are noted in the upper-right corner of each panel.

Figure 4c shows the composite precipitation for the TD, TY, and EC cases over the North Pacific. The distribution of EC differs significantly from that of TDs and TYs in that the peak is located northeast of the center, whereas those of TD and TY are concentric. Typhoon-turned-ECs tend to precipitate along stationary fronts or to the north of the EC, and the average precipitation maximum is located about 3.5 degree (~40 km) northeast of the EC center (Figure 4c). The relative location of the disaster area in Shimokita Peninsula to the center location of EC9 at peak 1 and peak 2 are also plotted in Figure 4c. Both locations are in the precipitation zone, and location to the center for peak 2 is at the maximum precipitation area.

The precipitation distribution of EC9 studied in this paper is a typical EC precipitation pattern, although the analysis periods are different. In the five TC precipitation categories (shown in Figure S1), averaged within a 500 km radius, the order of precipitation was TY > STS > TS > TD > EC; EC had the lowest precipitation on average, but the precipitation was concentrated northeast of the center, suggesting that moist mid- and lower-level air from the typhoon may have contributed to the precipitation.

Composites of precipitation for Term 1 and Term 2 are shown in Figure 5. In this case, Term 1 is stronger north of the center. This is consistent with Shimazu (1998) [5], who pointed out that the precipitation area of typhoons approaching Japan is biased toward the north. In Term 2, the precipitation is generally weaker than that of Term 1, but the precipitation extends to the northeast, which is similar to that shown in Figure 4c. Previous studies have pointed out that in the transforming period, much precipitation tends to be observed along a front [11,19,20]. These studies deal with the cases of warm front and suggested the importance of conditional symmetric instability [11,20]. Although there are differences in the front type (warm front/stationary front), area, and stages, there seems

to be a common character, namely, the maximum precipitation at the northeastern part of the ECs.

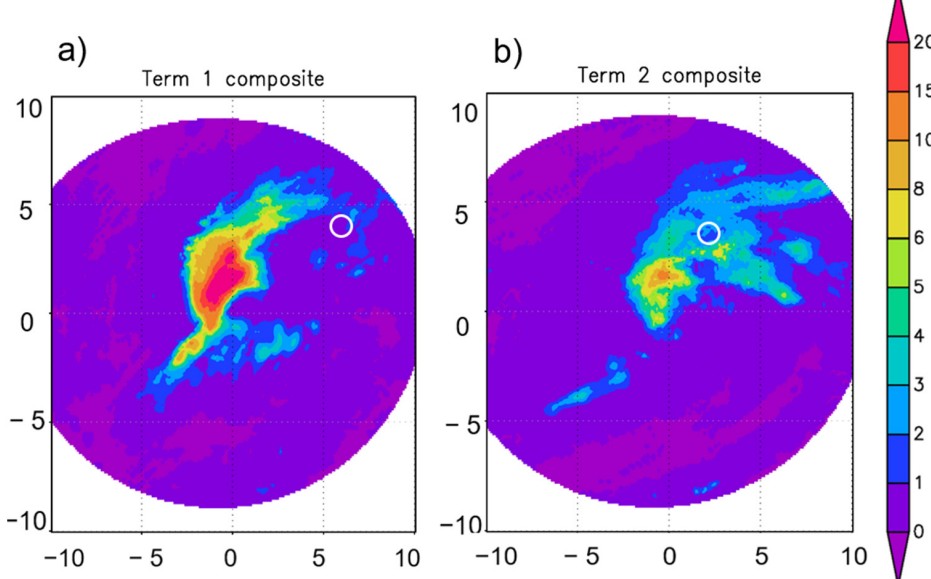

**Figure 5.** Composited precipitation distribution (mm/h) for (**a**) Term 1 (0–15 UTC of 9th) and (**b**) Term 2 (from 15 UTC of 9th to 6 UTC of 10th). The same scales are used for (**a**,**b**). While open circles indicate the relative locations of Shimokita Peninsula at peak 1 and peak 2.

Analysis of the moisture sources of heavy rainfall along the stationary front is an option for future research, but next we will show the relationship between large-scale moisture flow and the location of the center of EC9.

### 3.3. Large-Scale Moisture Transport and TC/EC Precipitation

Since precipitation along the EC9 front is thought to have caused the heavy rainfall, the amounts of vertically integrated specific humidity ($\frac{1}{g}\int_{p0}^{p1} q\,dp$) and moisture flux ($\frac{1}{g}\int_{p0}^{p1} q\,\vec{V}\,dp$) in the middle and lower levels by ERA5 are shown (Figure 6a–d). Moisture flux divergence ($\nabla \cdot q\vec{V}_{950}$) at 950 hPa by MSM are also shown (Figure 6e,f). At first glance, EC9 appears to have more moisture concentrated in the central part, especially at the middle level. It seems to retain characteristics of a typhoon, TC9.

There is a dry slot with little moisture in the outer part of the central moisture-abundant area at the middle level (Figure 6a,b). The tips of dry air passes over Shimokita between peak 1 (136° E, 40° N of Figure 6a) and peak 2 (140° E, 41° N of Figure 6b). This corresponds to a weakening of precipitation at 15 UTC (Figure 3b). The wind and specific humidity at 500, 700, and 900 hPa geopotential heights at 9 UTC, 15 UTC, and 18 UTC are shown in Figure S3.

Figure 6e,f show clear differences in the moisture convergence at a lower troposphere: strong precipitation and a convergence zone were observed along the stationary front at peak 2 and were identified as heavy rainfall in the Shimokita region. This heavy rainfall zone appeared at about 380 km northeast from the EC9 center (Figure 6f), which corresponds to a maximum precipitation zone of "typhoon-turned-EC".

Figure 6g–j represent vertically integrated moisture flux and its divergence ($\nabla \cdot (\frac{1}{g}\int_{p0}^{p1} q\,\vec{V}\,dp)$). The flux vectors are the same with that of Figure 6a–d, respectively. Strong moisture convergence (~20 mm/h (same with kg m$^{-2}$ h$^{-1}$)) can be found over the Shimokita Peninsula in a lower troposphere for both peak times (Figure 6i,j). In the middle troposphere (Figure 6g,h), moisture converges (~10 mm/h) over the Shimokita Peninsula.

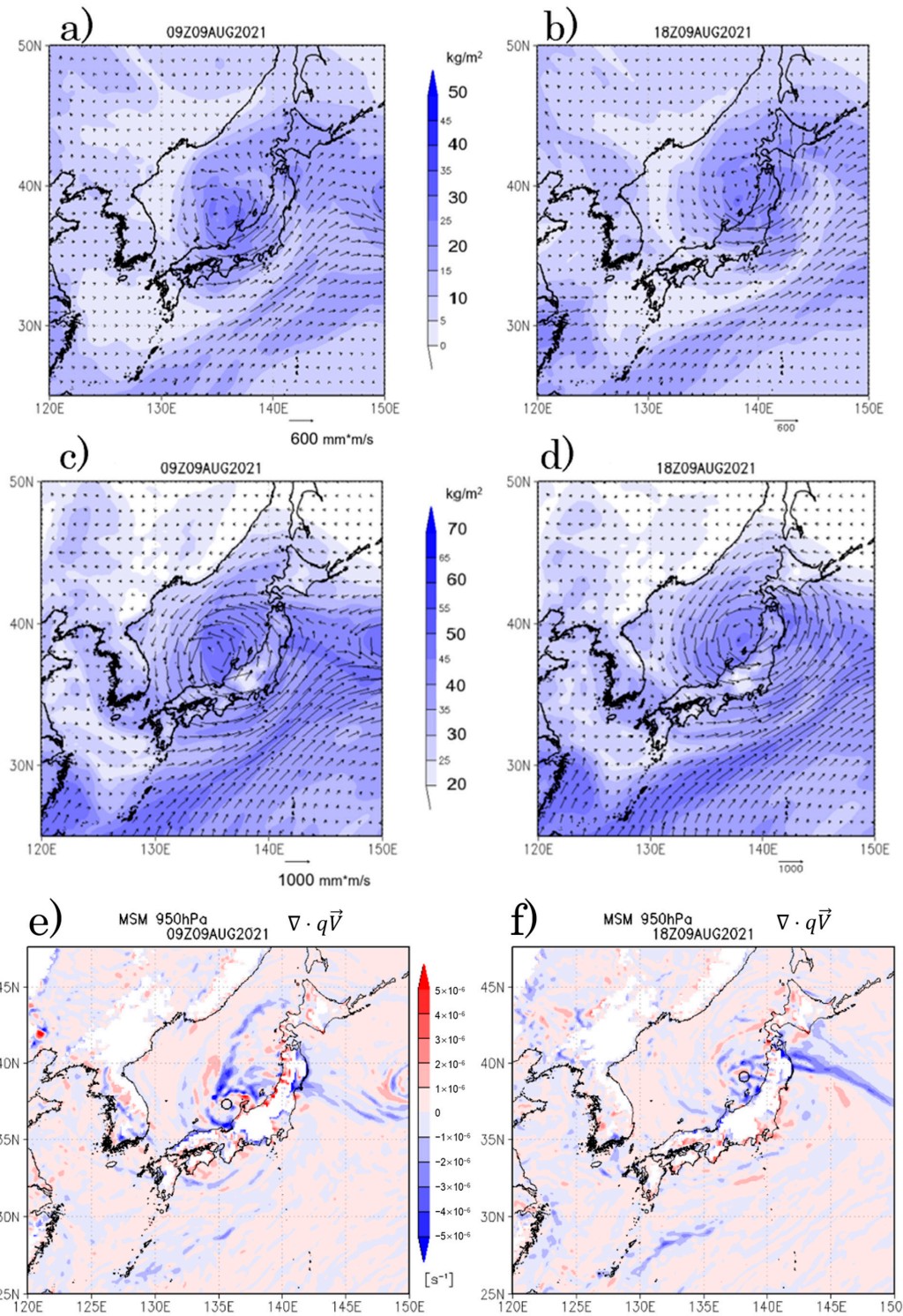

**Figure 6.** *Cont.*

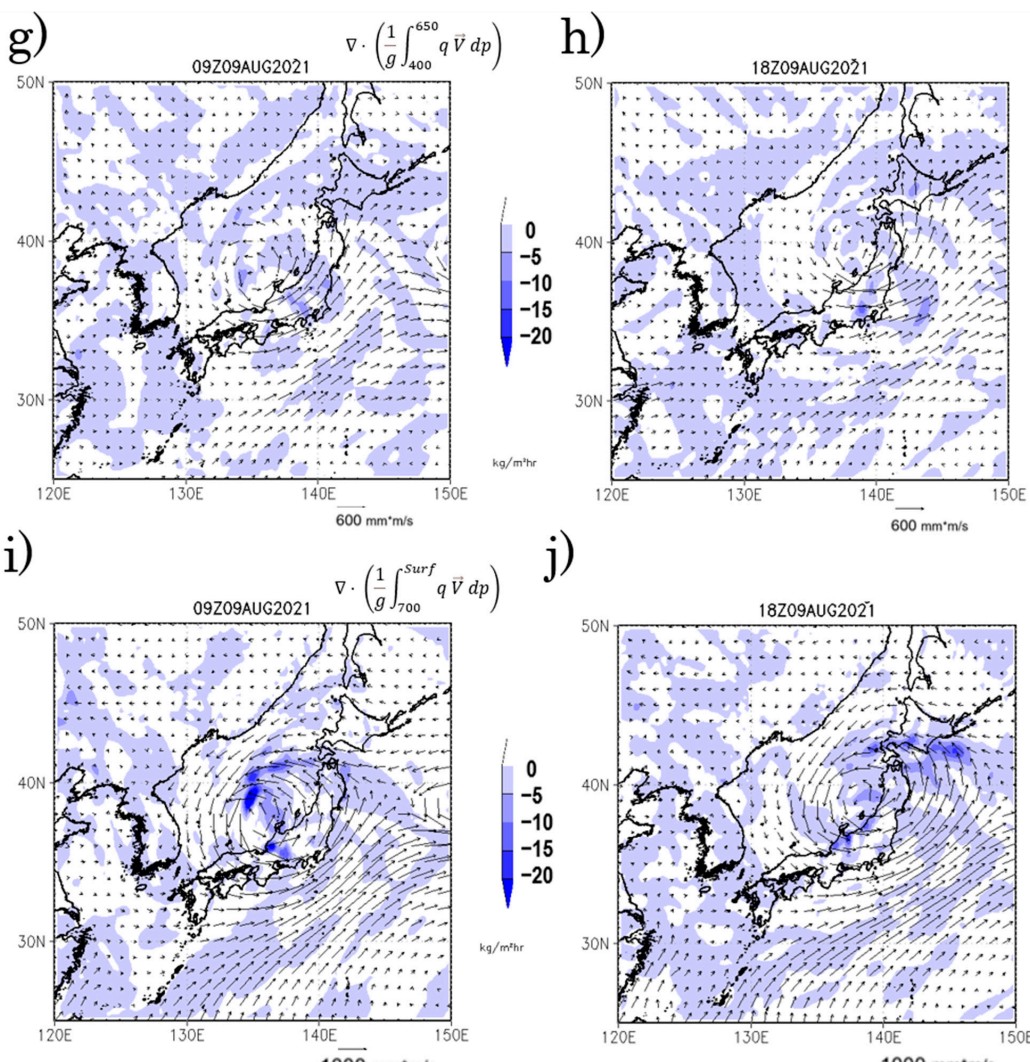

**Figure 6.** Vertically integrated specific humidity (kg/m²) (shade)and moisture flux (kg/m·s) (vector) in the middle troposphere (400–650 hPa) at (**a**) 9 UTC (peak 1) and (**b**) 18 UTC (peak 2) on 9 August. (**c**) Same as (**a**) but for a lower troposphere (700 hPa—surface). (**d**) Same as (**b**) but for a lower troposphere (700 hPa—surface). (**e**) Moisture divergence (blue means convergence) at 950 hPa at 9 UTC (unit: 1/s). (**f**) Same as (**e**) but for 18 UTC. Black open circles in (**e**,**f**) designate the location center of EC9. (**g**) Vertically integrated moisture flux (kg/M·s) (vector) and its divergence (kg/m²·h) in the middle troposphere (400–650 hPa). Only convergence (negative divergence) areas are shaded. (**h**) Same as (**g**) but for 18 UTC (peak 2). (**i**) Same as (**g**) but for a lower troposphere (700 hPa—surface). (**j**) Same as (**i**) but for 18 UTC (peak 2).

### 3.4. Vertical Distribution of Equivalent Potential Temperature and Moisture Transport

To establish the differences in moist–static energy transport, we compared the equivalent potential temperature ($\theta_e$) and wind fields at 950, 700, and 500 hPa heights for both peaks (at 9 UTC and 18 UTC) and intermittent weak precipitation time (at 15 UTC) in Figure 7. For peak 1, $\theta_e$ is high in the middle levels over the Shimokita Peninsula. A relatively low $\theta_e$ around 138 E/41 N corresponds to the dry slot shown in Figure 6a. While for peak 2, very high $\theta_e$ is transported from the south (Figure 7g). Additionally, in the lower level, high $\theta_e$ is seen in the southeastern part of Aomori Prefecture (Figure 7i), and it has a stronger north–south gradient, which is identified with a stationary front (Figure 7i), than that of peak 1 (Figure 7c). In between the two terms, at 15 UTC, $\theta_e$ is relatively low at 500 hPa (Figure 7d) and 700 hPa (Figure 7e) over the Shimokita Peninsula. At 12 UTC (shown in Figure S4), a clear minimum $\theta_e$ is observed at 400 hPa and 500 hPa heights. In the middle

level, a $\theta_e$ minimum to the west of the target area may cause a suppression of convection, which result in weaker precipitation over Shimokita compared to both of the peak times.

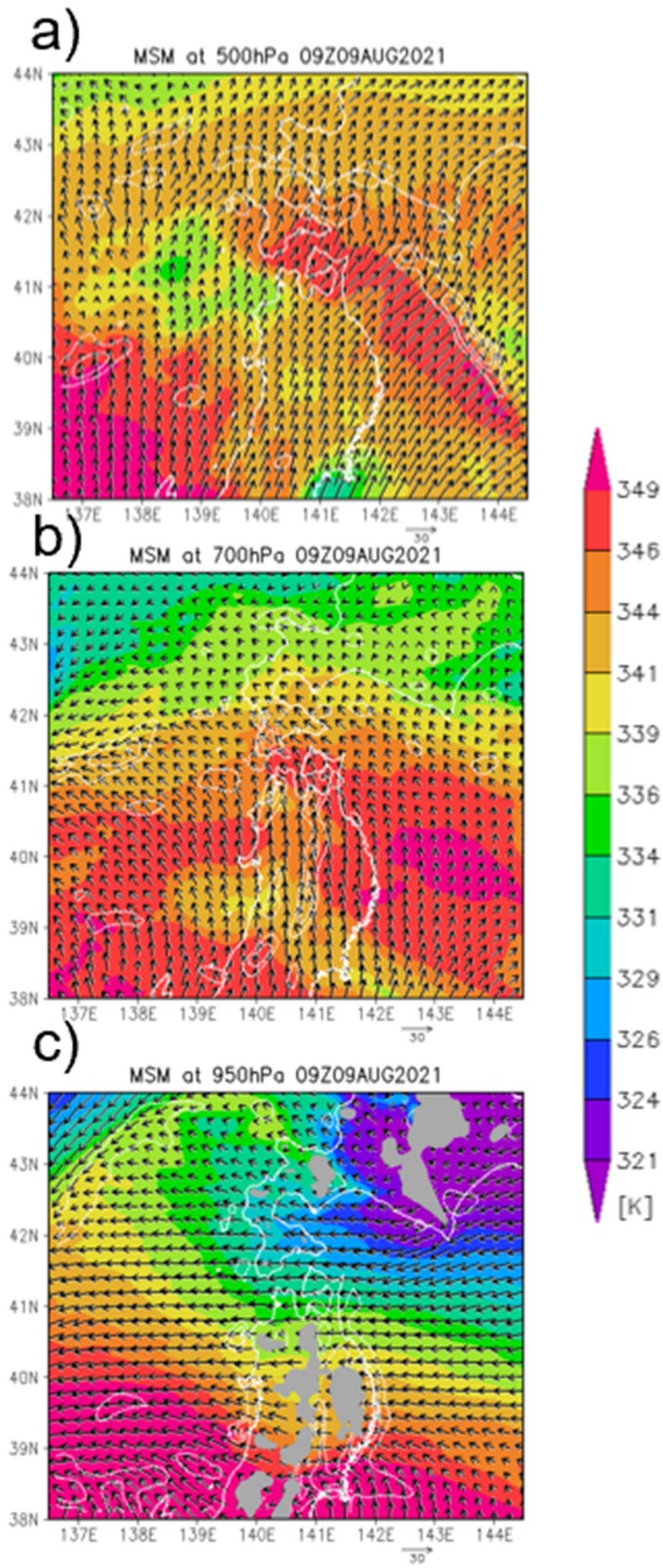

**Figure 7.** *Cont.*

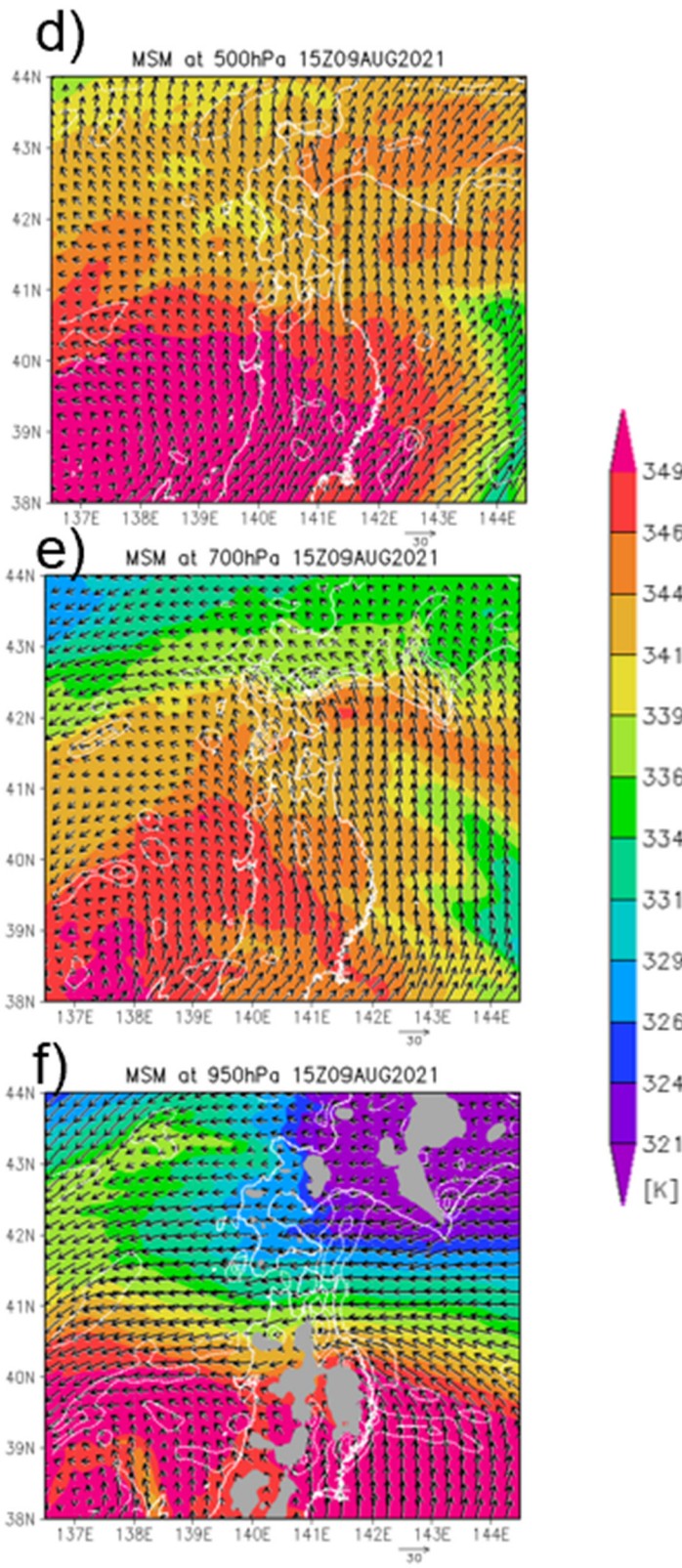

**Figure 7.** *Cont.*

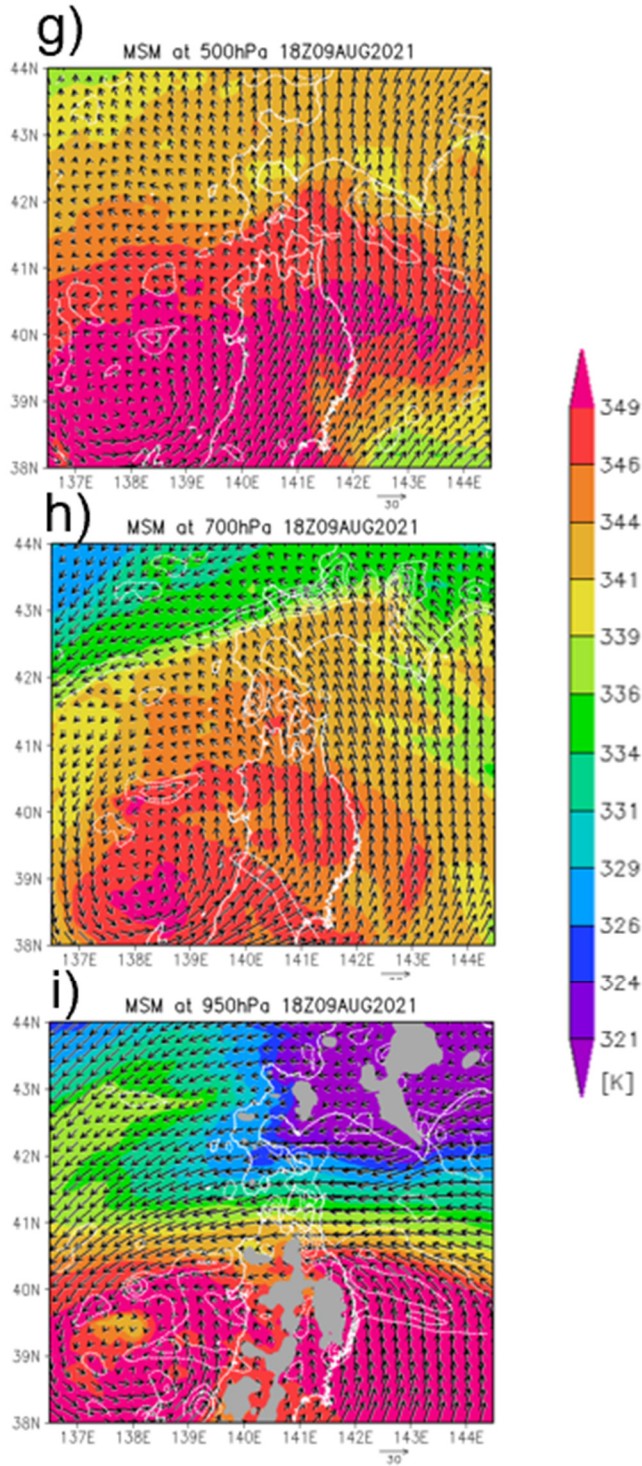

**Figure 7.** Equivalent potential temperature (θe, unit: K) and horizontal wind vectors calculated from the MSM. White dashed contours indicate convergence (contour interval is $10^{-4}$ s$^{-1}$). (**a**) The 500 hPa at 9 UTC 9 August 2021. (**b**) Same as (**a**) but for 700 hPa. (**c**) Same as (**a**) but for 950 hPa. Gray area in (**c**) indicates land areas. The color scale for equivalent potential temperatures and wind vectors are common for all 9 panels (**a–i**). (**d**) Same as (**a**) but for 15 UTC on 9 August 2021. (**e**) Same as (**d**) but for 700 hPa. (**f**) Same as (**d**) but for 950 hPa. Gray area in (**f**) indicates land areas. (**g**) Same as (**a**) but for 18 UTC on 9 August 2021. (**h**) Same as (**d**) but for 700 hPa. (**i**) Same as (**d**) but for 950 hPa. Gray area in (**i**) indicates land areas.

Then, we show the vertical cross section of moisture convergence along the Shimokita area. The area is indicated at Figure 1a with blue arrows. Figure 8a shows a north–south cross section of the zonal wind and the convergence of the moisture flux, and Figure 8c shows the same for the east–west cross section at peak 1. The same cross sections at intermittent times (12 UTC and 15 UTC) are shown in Figure S5. The same cross sections for peak 2 are shown in Figure 8e,g. Each vertical cross section of moisture flow is accompanied by the relevant cross section of 2 h of precipitation (Figure 8b,d,f,h).

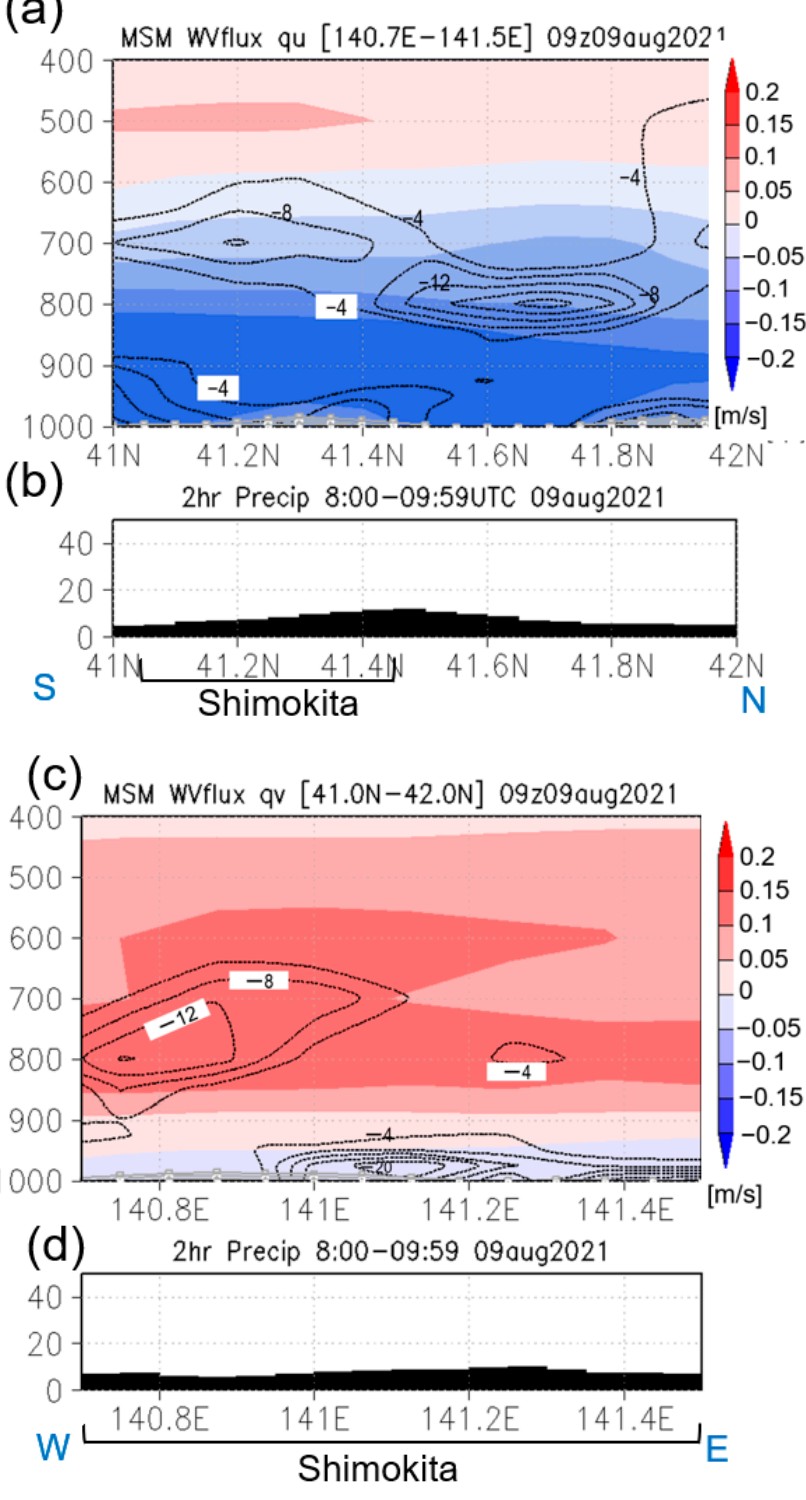

**Figure 8.** *Cont.*

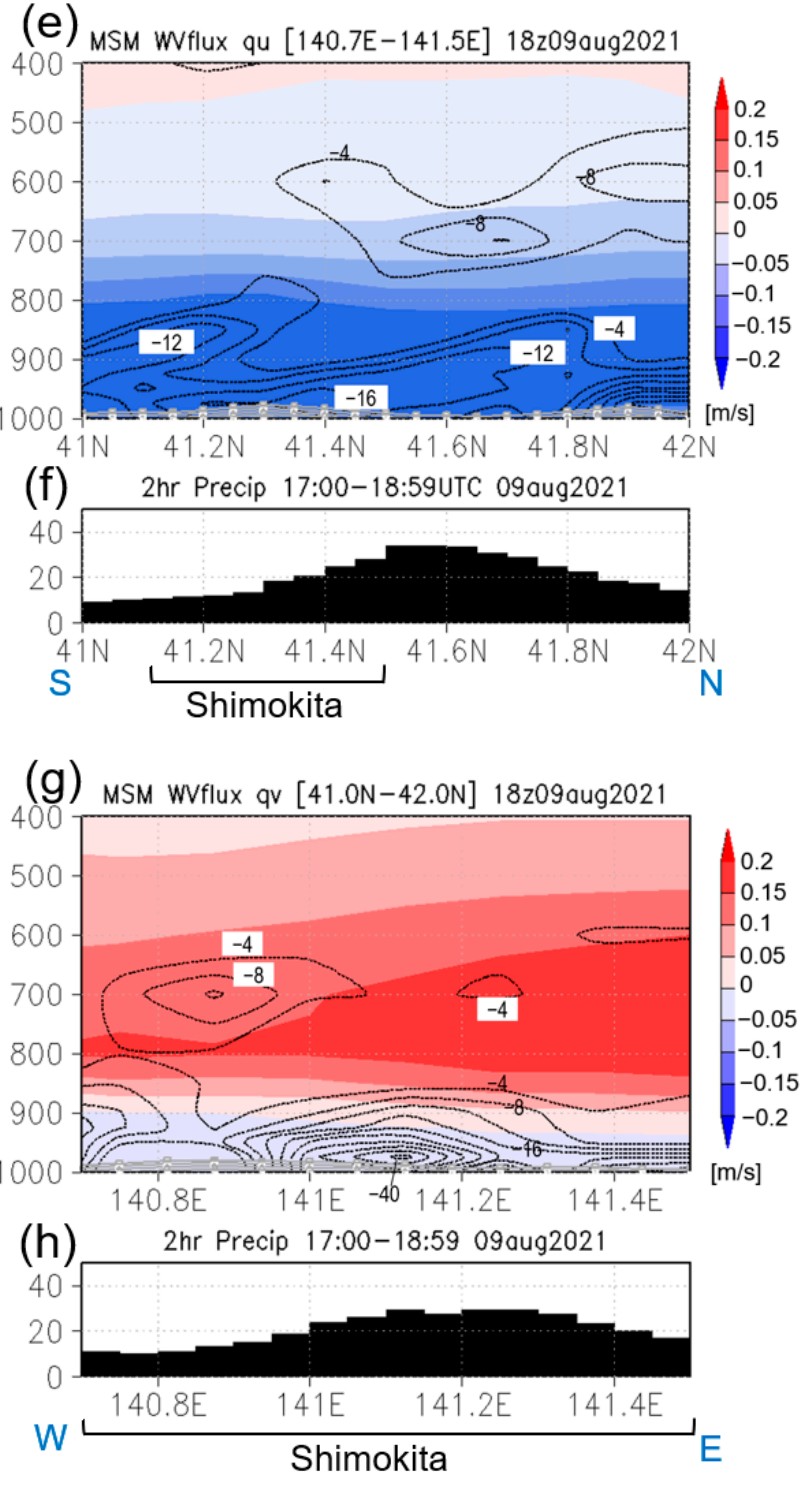

**Figure 8.** (**a**) Vertical cross section of eastward moisture flux (m/s) averaged over 140.70–141.5° E at 9 UTC 9 August. Red (blue) color indicates westerly (easterly). Contours indicate moisture convergence ($\times 10^{-8}$/s). (**b**) Cross section of precipitation (mm/2 h) averaged over 140.70–141.5° E during 8–10 UTC 9 August. (**c**) Vertical cross section of northward moisture flux (m/s) averaged over 41–42° N at 9 UTC 9 August. Red (blue) color indicates southerly (northerly). Contours indicate moisture convergence ($\times 10^{-8}$/s). (**d**) Cross section of precipitation (mm/2 h) averaged over 41–42° E at 9 UTC 9 August. (**e**) Same as (**a**) but for 18 UTC. (**f**) Same as (**b**) but for 17–19 UTC. (**g**) Same as (**c**) but for 18 UTC. (**h**) Same as (**d**) but for 17–19 UTC.

Throughout the periods, easterly prevails except for the middle level at peak 1 and upper levels. In peak 1 (Figure 8a), the zonal winds converge at 800 hPa over the Tsugaru Strait and achieve weak convergence at approximately 700 hPa over Shimokita Peninsula. Convergence also occurs over the land area in the lowest layer of the troposphere. For peak 2 (Figure 8e), however, convergence is strong in the entire lower troposphere, corresponding to a large amount of precipitation (Figure 8f). The difference between Term 1 and Term 2 is whether there is a convergence zone in the middle level over the Tsugaru Strait.

In contrast, the east–west cross sections (Figure 8c,g) show a reversal of the wind direction at around 900 hPa. Considering Figures 7 and 8 together, in both terms, at the stationary front (Figure 2c,d), cold air came from the north at the lower levels and a large amount of moisture supply came from the south at the middle and upper levels. A large amount of moisture entered the lower levels from the east (Figure 7c,f,i), but converged along the front to form precipitation. Both lower-level northerly to Shimokita and moisture convergence over there are stronger at peak 2 than that of peak 1.

The moisture convergence (shown in dashed contours in Figure 8a,c,e,g) of $1 \times 10^{-6}$ $(m^{-1}s^{-1})$ roughly corresponds to 3.7 mm/2 h if it integrates from the surface to 100 hPa. In peak 1, strong moisture convergences appeared over the Tsugaru Strait at 800 hPa (Figure 8a) and northerly converged at a lower level (Figure 8e). In peak 2, the moisture convergence along the stationary front (Figure 8g) is about 2~4 $\times 10^{-6}$ $(m^{-1}s^{-1})$, and it corresponds to 15~30 mm/2 h, which is almost the same amount of precipitation around the Shimokita Peninsula (Figure 8h). These strong convergences along the stationary front, namely, along the large gradient zone of $\theta_e$ in the lower troposphere, were very important for the torrential precipitation at the Shimokita event.

The role of lower-level easterly moisture supply should be considered with orography. The Tsugaru Strait, which easterly flow entered, is shaped like a corridor that is getting narrower. Due to this orography, wet and strong easterly wind tends to uplift, and convective clouds develop due to the moisture supply (air with high equivalent potential temperature) from the south. As with the heavy rains in Hiroshima and western Japan in 2014 and 2018 (Y19), the passage of warmer moist air through the Bungo Channel was one of the causes of torrential rains [13]. The interaction between large-scale moisture flow, the role of dry air, and local orographic enhancement is another option for future research.

A strong precipitation zone along the stagnant front, at 300–500 km northeast of EC9 center, was associated with the disastrous precipitation. A weather system that brings heavy rainfall ahead of the center of the EC—a stationary front—will arrive before the center of the EC; therefore, caution is needed. Our results are consistent with those of previous reports (JMA Aomori Local Office, 2021): the moist easterly winds in the lower troposphere rose and formed a precipitation zone (Term 1), and considerably warmer air in the middle and lower troposphere flowed into the target area from the south (Term 2). However, we emphasize that Term 2 has much more precipitation caused by the stationary front associated with EC9.

## 4. Conclusions

We aimed to clarify the quantitative structure of precipitation and moisture transport according to a typhoon-turned extratropical cyclone (EC9) in the Shimokita heavy rainfall event from 9–10 August 2021. We first developed intensive hourly grid precipitation data with rain gauges (APHRO_JPh) for the target period using the method described in Y19. We then compared the precipitation distribution of EC9 with composited TC precipitation of five years (2015–2019, 95 TCs). Then, for the Shimokita heavy rainfall event, we analyzed the vertical structure of the equivalent potential temperature and moisture flux convergence with precipitation according to EC9.

The main findings of this study are as follows:

- Using additional rain gauges from Aomori Prefecture and MLIT, in addition to JMA's AMeDAS, we successfully represented the heavy precipitation pattern and its time series over the landslide region.

- In most locations where the landslide disaster occurred, there were two precipitation peaks: at 9 UTC (Term 1: 0–15 UTC) and 18 UTC (Term 2: 15 UTC (9 August)—6 UTC (10 August)) on 9 August. The total precipitation over Shimokita Peninsula was greater in Term 2 than in Term 1.
- A common feature of the two terms was strong wind shear from the lower troposphere to the upper troposphere, with easterly winds in the lower troposphere below 900 hPa and warm moist air with a high equivalent potential temperature from the southeast flowing into the target area at approximately 850–700 hPa, and from the south flowing into the heavy rainfall area in the upper troposphere from 700 hPa.
- A strong gradient of equivalent potential temperature (especially for Term 2) and a northerly component with a lower troposphere and moisture convergence indicate the existence of the stationary front shown in the weather chart.
- During Term 1, moisture convergence was found over the Tsugaru Straits at approximately 800 hPa, while Term 2 did not show such a maximum.
- The large-scale moisture transport/convergence clarified that the heavy precipitation zone and strong moisture convergence that caused the Shimokita event were associated with the stationary front of EC9. The precipitation distribution, which has a precipitation peak northeast of the EC center, is a typical typhoon-turned-EC precipitation distribution.

**Supplementary Materials:** The following supporting information can be downloaded at https://www.mdpi.com/article/10.3390/atmos15010094/s1. Figure S1: TC precipitation composites; Figure S2: Areal averaged hourly precipitation; Figure S3: Wind and specific humidity from ERA5; Figure S4: Wind and equivalent potential temperature from MSM; Figure S5: Vertical cross section of moisture flux and precipitation at intermittent time.

**Author Contributions:** Conceptualization, A.Y. and S.S.; methodology, A.Y.; software, A.Y. and S.S.; validation, S.S.; formal analysis, A.Y. and S.S.; investigation, A.Y. and S.S.; resources, A.Y.; data curation, S.S. and A.Y.; writing—original draft preparation, S.S. and A.Y.; writing—review and editing, A.Y.; visualization, S.S. and A.Y.; supervision, A.Y.; project administration, A.Y.; funding acquisition, A.Y. All authors have read and agreed to the published version of the manuscript.

**Funding:** This research was funded by collaboration research of the Shimokita Geopark under Mutsu city. Part of this study was supported by 2021 Hirosaki University Institutional Fund.

**Institutional Review Board Statement:** Not applicable.

**Informed Consent Statement:** Not applicable.

**Data Availability Statement:** The APHRODITE and APHRO_JP precipitation data are openly available at http://aphrodite.st.hirosaki-u.ac.jp/. The original precipitation data and gridded data created in this study are not publicly available because data from Aomori Prefecture belongs to them and is not publicly available. The gridding algorithm used in this study adopts "the station value conservation (SVC) option", so that original values remain. However, gridded data created in this study can be available on request from the corresponding author by understanding related rights.

**Acknowledgments:** We thank Jin Kawashiro for data preparation and Yuno Sakashita and Ayano Shirakawa for providing composite charts on the tropical cyclones. We also thank Aomori local Meteorological Office, Japan Meteorological Agency and Aomori Prefectural Office for providing various meteorological data and documents related to the Shimokita event.

**Conflicts of Interest:** The authors declare no conflicts of interest.

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
