# Peer review of "Precipitation and Moisture Transport of the 2021 Shimokita Heavy Precipitation: A Transformed Extratropical Cyclone from Typhoon#9"

_atmosphere, doi:10.3390/atmos15010094_

Round 1
Reviewer 1 Report
Comments and Suggestions for Authors
GENERAL REMARKS:
Overall the manuscript provides worthwhile information and insights on the Shimokita heavy precipitation. Yet, I see a need to improve the readability of the manuscript and the discussion of the results.
The size of the figures is too small. Figure 1, 2 b-d, 3 acd, 5 a-f, 6 e-I (by far too small), 7 Please consider that the figures should be still readable when the pdf is printed.
‘Please improve the English style. E.g. 2.1 several times “was used” is used in a small paragraph. Also a technical description should be pleasant to read.
The authors know what they are doing and why, but it is partly not well explained to the reader. Some steps and their reasoning should be described in more detail. See detailed comments.
There is a lack of discussion of works by others. Please add, together with respective references. Currently too few references.
DETAILED COMMENTS:
Line 99 “APHRO_JPh was not created” Please consider to replace ít by “because respective measurements were not available”.
Section 2.2.:
It is now well described and motivated why 3 different models are used JMA-NHM, MSM (JMA) and ERA-5.
I guess “local moisture transport, equivalent potential temperature and wind 104 fields” could have been also taken from ERA-5. JMA-NHM has a high resolution but probably not the needed coverage ?
Please improve the description and motivation.
L 123: “with tropical cyclones [8] created in the laboratory is shown in Figure 4.”
It is not clear what this means, please describe in more detail. Further, why is this study needed. How can laboratory data be a reference ?
L149: Please explain in more detail what a “stagnation front” is and provide a reference.
L 164: “This seems 164 to have left behind the character of TC9”
Not clear what is meat. Please clarify.
4 Summay:
-Please change to Conclusion.
- The main findings are well presented, but there is a lack of discussion in the larger context of why this severe weather appeared. Weather is chaotic, thus it appeared to a certain extent by pure chance, but the probability of extreme events might have increased due to the climate change (e.g. on average increase of moisture, more energy means a higher probability to reach extremer states). Please discuss.
- I also miss a discussion of the effect of the large scale (global) circulation and/or teleconnections on the severe weather event. Please add.
Comments on the Quality of English LanguageSee detailed comments of my review.
Author Response
We intend to send the manuscript for English proof if needed. Detailed point to point reply, please see the attachment.

Reviewer 2 Report
Comments and Suggestions for Authors
The topic is interesting: focusing on the heavy local precipitation associated with an extra cyclone which transformed from a tropical cyclone. About this authors should indicate when the tropical cyclone transformed, how to identify and marked it in Figure 2a. Besides, some comments are following:
Major comments
1. The logic of the article is quite chaotic, such as after analyzing Fig.5(L160-175), then Fig.4 again (L186-197) and so on
2. The Figure 5 title did not correspond to that shown in Figure 5. The unit of divergence qV shown in Fig.5e is wrong.
3. Figure 6 is not very clear, needs to be improved.
4. : What did the contour and shading in Figure 7 represent, respectively? The unit of flux is not right. The quality of Figure 7 needs to be improved.
5. Figure should not be put under the title "introduction"
Minor comments
1. What are ‘the latter peak ‘ in L20 and ‘the second peak’ in L22, respectively?
2. In Fig.2a, it is better to mark the time when the EC9 made deformation
3. L137 , removing a “The”
Comments on the Quality of English LanguageWriting can be improved, such as it is difficult to understand the statement in L295-299 . Three points is OK in summary. One is the heavy precipitation characteristics, the second one is direct factors (such as moisture flux and convergence) which are associated with heavy precipitation. the third one is the large-scale which affected direct factors.
Author Response

(The authors gave the same response as above.)

Round 2
Reviewer 2 Report
Comments and Suggestions for Authors
Major comments
1. The writing in the text is to some extent contradictory, such as L17 “easterly winds in the lower troposphere”, L19 “a northerly in the lower troposphere”. The presentation in the text needs improvement, especially the abstract and conclusion.
2. Basic theoretical knowledge needs to be strengthened. Such as moisture flux ( vector or qu zonal direction or qv meridional direction) is kg m-1s-1 , it is simple to know what the unit of convergence of moisture flux (
) . So, the units shown in Fig.6e, 6f, Fig.8 are all wrong.

The writing needs improvement to enhance readability. In addition to logical aspect, refinement is also needed.
Author Response
Dear Reviewer,
Thank you very much for careful reading and comments.
Our response is written in blue in this form, and modified (added) part is written in red in the text.
I discussed with editorial office about English native check. We have sent the original manuscript for English proofing; however, it seems not to be perfect. Now, it is possible to send it to another company, but I am afraid it takes another week. Ms. Melisa Lei explained that MDPI has English proofing after acceptance. So I prefer to use this option.
Major comments
- The writing in the text is to some extent contradictory, such as L17 “easterly winds in the lower troposphere”, L19 “a northerly in the lower troposphere”. The presentation in the text needs improvement.
Thank you very much for pointing out this. We found description in L17 is for the first peak, and that of L19 is for the 2nd (latter) peak. In order to clarify this, the abstract words exceeded the limit (200 words). Hence, we simplify the abstract to meet the limit.
- Basic theoretical knowledge needs to be strengthened. Such as moisture flux ( )vector or qu zonal direction or qv meridional direction) is kg m-1s-1 , it is simple to know what the unit of convergence of moisture flux ( ) . So, the units shown in Fig.6e, 6f, 8 are all wrong.
Thank you very much for kind explanation. We carefully checked all computational scheme and units. We showed (calculated) two moisture fluxes. at each pressure level and vertically integrated moisture flux ( . The latter, namely, the unit of is kg m-1s-1 , as reviewer pointed out. So, the unit of vertical integrated divergence/convergence of moisture flux is kg m-2s-1. However, moisture flux at each level has different unit. (q: specific humidity has no unit (kg/kg), and unit of (u,v) is m s-1. To clarify this, we added explanation in section 2.2, and added figure and their explanation for the Figure 6 (Fig.6g-6j).
